# Hierarchical Variational Memory for Few-shot Learning Across Domains

**Yingjun Du[1], Xiantong Zhen[1,2], Ling Shao[3], Cees G. M. Snoek[1]**
[1]AIM Lab, University of Amsterdam [2]Inception Institute of Artificial Intelligence
[3]National Center for Artificial Intelligence, Saudi Data and Artificial Intelligence Authority

## Abstract

Neural memory enables fast adaptation to new tasks with just a few training samples. Existing memory models store features only from the single last layer, which does not generalize well in presence of a domain shift between training and test distributions. Rather than relying on a flat memory, we propose a hierarchical alternative that stores features at different semantic levels. We introduce a hierarchical prototype model, where each level of the prototype fetches corresponding information from the hierarchical memory. The model is endowed with the ability to flexibly rely on features at different semantic levels if the domain shift circumstances so demand. We meta-learn the model by a newly derived hierarchical variational inference framework, where hierarchical memory and prototypes are jointly optimized. To explore and exploit the importance of different semantic levels, we further propose to learn the weights associated with the prototype at each level in a data-driven way, which enables the model to adaptively choose the most generalizable features. We conduct thorough ablation studies to demonstrate the effectiveness of each component in our model. The new state-of-the-art performance on cross-domain and competitive performance on traditional few-shot classification further substantiates the benefit of hierarchical variational memory.

## 1 Introduction

Few-shot learning with an external memory is known to learn new concepts quickly with only a few samples, especially when embedded in a meta-learning setting (Santoro et al., 2016). A common tactic is to store short-term memory (Munkhdalai & Yu, 2017; Munkhdalai et al., 2018; Kaiser et al., 2017) as obtained from the support set of the current task, and to empty it at the end of a task. Another tactic is to let the memory store long-term knowledge distilled from all the training tasks (Zhen et al., 2020a), which provides a conceptual context to learn a new task. The long-term memory has demonstrated effectiveness in enabling a model to quickly learn new few-shot learning tasks within domains (Zhen et al., 2020a). Nonetheless, for both the short- and long-term memory tactics, the learning ability is limited when facing a task from an unseen domain (Tseng et al., 2020; Guo et al., 2020; Cai & Shen, 2020; Du et al., 2021).

Cross-domain few-shot learning (Tseng et al., 2020; Guo et al., 2020) is even more challenging than the conventional few-shot learning problem as it also has to take distribution shift into account. In cross-domain few-shot learning, directly using traditional external memory, which often only stores high-level semantic features (Zhen et al., 2020a), is unable to find the suitable semantics in memory due to the domain shift causing side effects. Nonetheless, low-level features like textures, shapes, edges, etc. from training domains are often still meaningful for a new domain (Adler et al., 2020). We hypothesize, features at different levels would play distinctive roles in generalizing across domains. It remains unexplored to investigate hierarchical features in memory that can enable quick adaptation, as humans do, when faced with a few samples from new domains.

In this paper, we make three contributions. First, we propose hierarchical variational memory, which accrues and stores the long-term semantic knowledge of multiple network layers from past experiences. Each entry stores features at different hierarchical levels by summarizing the feature representations of class samples, thus generalizing well in presence of a domain shift between the training and test distributions. We formulate the memory recall at different hierarchical levels as a

variational inference of the latent memory, which is an intermediate stochastic variable. Second, to better deploy the hierarchical memory, we introduce a hierarchical variational prototype model to obtain prototypes at different levels as well. We also develop the hierarchical variational prototype in a latent model by treating the prototypes at different levels as the latent variable. In doing so, the model is endowed with the ability to flexibly rely on low-level textures or high-level semantic features if the domain shift circumstances so demand. As the third contribution, we propose to learn the weights associated with the prototype at each level in a data-driven way, which can explore and exploit the importance of the different feature levels for domain generalization. We leverage the few training samples in the new domain to generate reasonable weights for each prototype, since these samples already carry sufficient appearance cues about the new domain, which enables the model to adaptively choose the most generalizable features. Two different hierarchical variational inference problems are included in our optimization objective: 1) the inference of the latent memory, which conditions on the latent memory of previous layers and the memory contents of the current layer; 2) the inference of prototypes, which treats the support set, the latent memory of the current layer and the prototype of previous layers as conditions.

We demonstrate the effectiveness of the proposed hierarchical memory by extensive experiments on the cross-domain few-shot learning classification tasks introduced by (Guo et al., 2020) and the more traditional within-domain few-shot learning classification tasks (Snell et al., 2017). We conduct an extensive ablation study to demonstrate the contributions of different components in our model. We also analyze how hierarchical memory adaptively chooses the weights in the face of domain change and identifies the appropriate layer of the prototype that should be used. An extensive evaluation of cross-domain few-shot and traditional few-shot classification benchmarks reveals that our hierarchical memory consistently achieves results that are better, or at least competitive, compared to other methods.

## 2 METHODOLOGY

### 2.1 PRELIMINARIES

In the following, we present the relevant background on few-shot classification and its cross-domain setting, as well as the preliminaries of the (variational) prototypical network and variational semantic memory (Zhen et al., 2020a).

**Few-shot classification** In the few-shot classification scenario, we define the $N$-way $K$-shot classification problem, which is comprised of the support sets $S$ and query set $\mathcal{Q}$. Each task also called an episode, is a classification problem sampled from a task distribution $p(\mathcal{T})$. The 'way' of the episode refers to the number of classes in the support, while the 'shot' of the episode refers to the number of examples per class. Tasks are drawn from a dataset by randomly sampling a subset of classes, sampling points from these classes, and then partitioning the points into support and query sets. Episodic optimization (Vinyals et al., 2016) iteratively trains the model by taking one episode update at a time.

**Cross-domain setting** For few-shot classification, the test tasks are typically assumed to come from the same task distribution $p(\mathcal{T})$ as the training tasks. The recently introduced task of cross-domain few-shot learning (Guo et al., 2020) considers the few-shot classification challenge under domain shift. Concretely, the training task comes from a single source domain $p(\mathcal{T}_0)$, while the test tasks come from several unseen domains $\{\mathcal{T}_1, \cdots, \mathcal{T}_M\}$. The goal of cross-domain few-shot learning is to learn a meta-learning model using a single source domain that generalizes to several unseen domains.

**Prototypical network** Our model builds upon the prototypical network (Snell et al., 2017), which is widely used for few-shot image classification. It computes a prototype $\mathbf{z}_k = \frac{1}{K}\sum_k f_\theta(\mathbf{x}_k)$ for each class through an embedding function $f_\theta$, which is realized by neural networks. It computes a distribution over classes for a query sample $\mathbf{x}$ given a distance function $d(\cdot, \cdot)$ as the softmax over its distances to the prototypes in the embedding space:

$$p(\mathbf{y}_i = k|\mathbf{x}) = \frac{\exp(-d(f_\theta(\mathbf{x}), \mathbf{z}_k))}{\sum_{k'} \exp(-d(f_\theta(\mathbf{x}), \mathbf{z}_{k'}))} \tag{1}$$

**Variational prototype network** The variational prototype network (Zhen et al., 2020a) is a powerful model for learning latent representations from small amounts of data, where the prototype $\mathbf{z}$ per class

is treated as a distribution. Given a task with a support set $S$ and query set $Q$, the ELBO takes the following form:

$$\log \big[ \prod_{i=1}^{|Q|} p(\mathbf{y}_i|\mathbf{x}_i) \big] \geq \sum_{i=1}^{|Q|} \Big[ \mathbb{E}_{q(\mathbf{z}|S)} \big[ \log p(\mathbf{y}_i|\mathbf{x}_i, \mathbf{z}) \big] - D_{\mathrm{KL}}(q(\mathbf{z}|S)||p(\mathbf{z}|\mathbf{x}_i)) \Big], \tag{2}$$

where $q(\mathbf{z}|S)$ is the variational posterior over $\mathbf{z}$, $p(\mathbf{z}|\mathbf{x}_i)$ is the prior, and $L_\mathbf{z}$ is the number of Monte Carlo samples for $\mathbf{z}$.

**Variational semantic memory** Variational semantic memory (Zhen et al., 2020a) is proposed to accumulate and store the semantic information from previous tasks for the inference of prototypes of new tasks. It consists of two main processes: *memory recall*, which retrieves relevant information that fits with specific tasks based on the support set of the current task; *memory update*, which effectively collects new information from the task and gradually consolidates the semantic knowledge in the memory. Variational semantic memory formulates the memory recall as a variational inference of the latent memory, which is an intermediate stochastic variable. The objective is written as follows:

$$\mathcal{L}_{\mathrm{VSM}} = \sum_{i=1}^{|Q|} \Big[ - \mathbb{E}_{q(\mathbf{z}|S,\mathbf{m})} \big[ \log p(\mathbf{y}_i|\mathbf{x}_i, \mathbf{z}) \big] + D_{\mathrm{KL}} \big[ q(\mathbf{z}|S, \mathbf{m})||p(\mathbf{z}|\mathbf{x}_i) \big]$$
$$+ D_{\mathrm{KL}} \big[ \sum_{i}^{|M|} \gamma_i p(\mathbf{m}|M_i)||p(\mathbf{m}|S) \big] \Big], \tag{3}$$

where $p(\mathbf{m}|S)$ is the introduced prior over latent memory $\mathbf{m}$ and $M$ denotes the memory content.

The above networks and memory only focus on the problem of few-shot classification within domain and their performance would be degrade for cross domain few-shot learning, because of the domain shift problem. In this paper, we propose hierarchical memory to address few-shot classification under domain shift.

## 2.2 HIERARCHICAL VARIATIONAL PROTOTYPE

In the variational prototype network (Zhen et al., 2020a), the probabilistic prototype is obtained by feeding the last layer of features to an amortization network. Hence, it is not possible to obtain the prototypes of different layers when addressing the domain shift problem, making it relying on the high-level prototype only, which is not necessarily shared between domains.

We introduce the hierarchical variational prototype network to generate different layer prototypes. Compared to its flat counterpart, it introduces $q(\mathbf{z}^l|\mathbf{z}^{l-1}, S)$ instead of $q(\mathbf{z}|S)$, so we approximate the true posterior by minimizing the KL divergence:

$$D_{\mathrm{KL}}[q(\mathbf{z}^l|\mathbf{z}^{l-1}, S)||p(\mathbf{z}^l|\mathbf{x}, \mathbf{y})], \tag{4}$$

where $\mathbf{z}^l$ is the prototype of layer $l$. By applying the Baye's rule, we then obtain the following ELBO:

$$\log \big[ \prod_{i=1}^{|Q|} p(\mathbf{y}_i|\mathbf{x}_i) \big] \geq \sum_{i=1}^{|Q|} \Big[ \big[ \frac{1}{L} \sum_{l=1}^{L} \mathbb{E}_{q(\mathbf{z}^l|S)} \big[ \log p(\mathbf{y}_i^l|\mathbf{x}_i, \mathbf{z}^l) \big] - D_{\mathrm{KL}}(q(\mathbf{z}^l|\mathbf{z}^{l-1}, S)||p(\mathbf{z}^l|\mathbf{z}^{l-1}, \mathbf{x}_i)) \big] \Big]. \tag{5}$$

In practice, the variational posterior $q(\mathbf{z}^l|\mathbf{z}^{l-1}, S)$ is implemented by an amortization network (Kingma & Welling, 2013) that takes the concatenation of the average feature representations of samples in the support set $S$ and the upper layer prototype $\mathbf{z}^{l-1}$ and returns the mean and variance of the current layer prototype $\mathbf{z}^l$. The hierarchical probabilistic prototype provides both a more informative representation than the deterministic prototype and the ability to capture different representation levels, making it more suitable for cross-domain few-shot learning. More importantly, the hierarchical variational prototype also provides a principled way of incorporating the prior knowledge of the different levels from the past experienced tasks. Therefore, we introduce the external hierarchical memory to enhance the prototype at different levels.

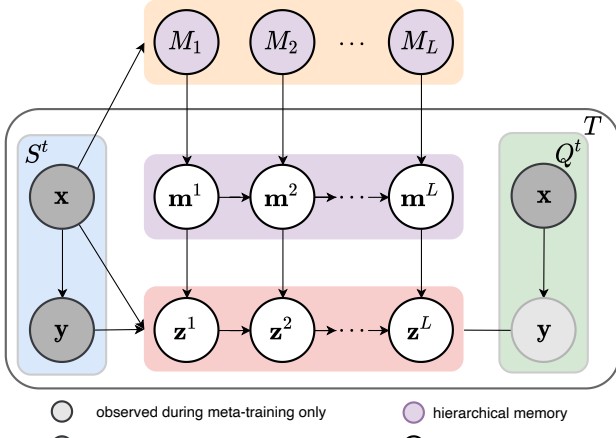

Figure 1: Computational graph of hierarchical variational memory for prototypical few-shot learning. The memory consists of $L$ layers of the hierarchy: each layer $M^l$ is used to build up the corresponding latent prototype $\mathbf{z}^l$. $S^T$ and $Q^t$ are the support set and the query set. $\mathbf{x}$ and $\mathbf{y}$ are the sample and label. The latent memory $\mathbf{m}^l$ is inferred from the memory content $M^l$ and the upper latent memory $\mathbf{m}^{l-1}$, and the latent prototype $\mathbf{z}^l$ is constructed from the support set $S$, latent memory $\mathbf{m}^l$, and the latent prototype $\mathbf{z}^{l-1}$ in the upper layer.

## 2.3 HIERARCHICAL VARIATIONAL MEMORY

We introduce the hierarchical variational memory to accrue and store the long-term semantic knowledge at different levels from previously experienced tasks for the hierarchical inference of prototypes for new tasks. By seeing more examples of objects at different semantic feature levels, the knowledge in the memory is enriched and consolidated, which allows for quick learning on new tasks.

Concretely, our model has $L$ layers of memory, each having $M$ storage units which store a key-value pair in each row of the memory array. The key in each memory stores the average feature representations of images from the same class and the value is the corresponding label. We adopt a similar memory mechanism as (Zhen et al., 2020a): memory recall and memory update, but we introduce a hierarchical version of memory recall.

We apply hierarchical variational memory into the hierarchical variational prototype network to generate the prototypes at different layers. From a Bayesian perspective, the prototype posterior can be inferred by:

$$q(\mathbf{z}^l|S) = \int q(\mathbf{z}^l|\mathbf{m}^l, \mathbf{z}^{l-1}, S)p(\mathbf{m}^l|\mathbf{z}^{l-1}, S)d\mathbf{m}^l, \qquad (6)$$

where $\mathbf{m}^l$ is the latent memory of layer $l$. Different from the variational semantic memory (Zhen et al., 2020a), we design a hierarchical variational approximation $q(\mathbf{m}^l|\mathbf{m}^{l-1}, M^l, S)$ to the posterior over the latent memory $\mathbf{m}^l$ of layer $l$ by inferring from $M^l$ conditioned on $S$. The hierarchical variational inference of prototypes by a hierarchical Bayesian framework is written as :

$$\tilde{q}(\mathbf{z}^l|M, S) = \sum_{a=1}^{|M|} p(a|M, S) \int q(\mathbf{z}^l|\mathbf{z}^{l-1}, S, \mathbf{m}^l)p(\mathbf{m}^l|M_a, \mathbf{m}^{l-1}, S)d\mathbf{m}^l, \qquad (7)$$

where $a$ is the addressed categorical variable, $M_a$ represents the corresponding memory content at address $a$, and $|M|$ indicates the memory size. We first leverage the support set $S$ and memory $M$ to generate the addressed categorical variable $a$, and then use the fetched memory content $M_a$, support set $S$, and the latent memory $\mathbf{m}^{l-1}$ of upper layer $l-1$ to infer the latent memory $\mathbf{m}^l$ of layer $l$. For the prototype $\mathbf{z}^l$, we use the upper layer prototype $\mathbf{z}^{l-1}$, the support set $S$, and the latent memory $\mathbf{m}^l$ as conditions.

By combining the ELBO in Eq. (5) for the hierarchical variational network and Eq. 7, we obtain the following ELBO for hierarchical variational inference:

$$\log\Big[\prod_{i=1}^{|Q|} p(\mathbf{y}_i|\mathbf{x}_i)\Big] \geq \sum_{i=1}^{|Q|}\left[\frac{1}{L}\sum_{l=1}^{L}\Big[\mathbb{E}_{q(\mathbf{z}^l|S, \mathbf{m}^l, \mathbf{z}^{l-1})}\big[\log p(\mathbf{y}_i^l|\mathbf{x}_i, \mathbf{z}^l)\big]\right.$$

$$\left. - D_{\mathrm{KL}}\big[q(\mathbf{z}^l|S, \mathbf{m}^l, \mathbf{z}^{l-1})||p(\mathbf{z}^l|\mathbf{z}^{l-1}, \mathbf{x}_i)\big] - D_{\mathrm{KL}}\Big[\sum_{i}^{|M|} p(\mathbf{m}^l|\mathbf{m}^{l-1}, M_i^l)||p(\mathbf{m}^l|\mathbf{m}^{l-1}, S)\Big]\Big]\right]$$

$$(8)$$

The overall computational graph of our approach is shown in Fig. 1. Directly optimizing the above objective does not consider the role of the different feature levels for domain generalization. Thus, we propose to learn the weights associated with the prototype at each level to achieve adaptive prototypes, which enables the model to adaptively choose the most generalizable features.

## 2.4 LEARNING TO WEIGH PROTOTYPES

Features from different layers contain different levels of semantic abstraction about an object and should contribute differently to the generalization of new tasks. Therefore, the importance of established prototypes from different feature levels need to be treated distinctively by depending on the domain from which a new task originates. To this end, we propose learning to weigh prototypes of different levels in a data-driven way.

Assume we have $L$ prototypes: $\{\mathbf{z}^1, \cdots, \mathbf{z}^L\}$, and we can obtain $L$ different prediction logits $\{\hat{\mathbf{y}}^1, \cdots, \hat{\mathbf{y}}^L\}$ based on Eq. (1). We can produce the final logit $\hat{\mathbf{y}}$ by bagging (Breiman, 1996) to vote which category it should be categorized into. However, this approach does not allow the model to adaptively choose the weight that different level prototypes should occupy based on the new domain.

In general, the final logit can be represented as a weighted sum: $\hat{\mathbf{y}} = \sum_{l=1}^{L} \alpha_l \hat{\mathbf{y}}^l$, where we learn the weight $\alpha_l$ for each prototype, by conditioning on the support set of the task. For cross-domain few-shot learning, we have a few samples $S$ when testing on new domains, we can generate a reasonable weight for prototypes at different levels of $S$, because, intuitively, these samples already carry sufficient domain information. To generate the weight $\alpha_l$ of layer $l$, we deploy a hypernetwork $f_\alpha^l(\cdot)$ that takes the average gradients (Munkhdalai et al., 2018) of all the support sets $S$ as input, and returns the weight $\alpha_l$ by softmax:

$$\alpha_l = \mathrm{softmax}(f_\alpha^l(\nabla_S^l)), \tag{9}$$

where $\nabla_S^l$ are the average gradients on layer $l$ of all the support sets. The hypernetwork $f_\alpha^l(\cdot)$ is first learned at meta-training time and then directly used as the support set $S$ from the test domain at meta-test time. Note that on the test domain we do not learn the parameter of $f_\alpha^l(\cdot)$; instead, we simply rely on the support gradient of each layer to generate its weight for the final result. We propose to learn the weights associated with the prototype at each level of the hierarchical variational prototype and hierarchical variational memory to adaptively unify different prototypes at different levels, which enables the model to adaptively choose the most generalizable features, thus achieving generalization for cross-domain few-shot learning.

## 3 RELATED WORKS

**Memory** Several works augment neural networks with an external memory module to improve learning (Santoro et al., 2016; Pritzel et al., 2017; Weston et al., 2014; Graves et al., 2016; Kaiser et al., 2017; Munkhdalai & Yu, 2017; Munkhdalai et al., 2018; Ramalho & Garnelo, 2019). (Santoro et al., 2016) equip neural networks with a neural Turing machine for few-shot learning. The external memory stores samples in the support set in order to quickly encode and retrieve information from new tasks. (Bornschein et al., 2017) proposed augmenting generative models with an external memory, where the chosen address is treated as a latent variable. Sampled entries from training data are stored in the form of raw pixels in the memory for few-shot generative learning. In order to store a minimal amount of data, (Ramalho & Garnelo, 2019) proposed a surprise-based memory module, which deploys a memory controller to select minimal samples to write into the memory. An external memory was introduced to enhance recurrent neural network in (Munkhdalai et al., 2019), in which memory is conceptualized as an adaptable function and implemented as a deep neural network. Semantic memory has recently been introduced by (Zhen et al., 2020a) for few-shot learning to enhance prototypical representations of objects, where memory recall is cast as a variational inference problem. In contrast to (Zhen et al., 2020a), our memory adopts a hierarchical structure, which stores the knowledge at different levels instead of the last layer only.

**Prototypes** The prototypical network for few-shot learning is first proposed by (Snell et al., 2017). It learns to project the samples into a common metric space where classification is conducted by computing the distance from query samples to prototypes of classes. Infinite Mixture Prototypes was

proposed by (Allen et al., 2019), which improved the prototypical network by an infinite mixture of prototypes that represents each category of objects by multiple clusters. (Triantafillou et al., 2020) combines the complementary strengths of Prototypical Networks and MAML (Finn et al., 2017) by leveraging their effective inductive bias and flexible adaptation mechanism for few-shot learning. (Zhen et al., 2020a) proposed the variational prototypical network to improve the prototypical network by probabilistic modeling of prototypes, which provided more informative representations of classes. Different from their work, we propose hierarchical variational prototype networks to generate different layer prototypes instead of using one prototype, which could capture information at different levels for few-shot learning across domains.

**Meta-learning** or learning to learn (Schmidhuber, 1987; Bengio et al., 1991; Thrun & Pratt, 1998), is a learning paradigm where a model is trained on distribution of tasks so as to enable rapid learning on new tasks. Meta-learning approaches to few-shot learning (Vinyals et al., 2016; Ravi & Larochelle, 2017; Snell et al., 2017; Zhen et al., 2020a; Finn et al., 2017; Du et al., 2021) and the domain generalization (Du et al., 2020; Xiao et al., 2021; 2022) differ in the way they acquire inductive biases and adapt to individual tasks. They can be roughly categorized into three groups. Those based on metric-learning generally learn a shared embedding space in which query images are matched to support images for classification (Vinyals et al., 2016; Snell et al., 2017; Santoro et al., 2016; Oreshkin et al., 2018; Allen et al., 2019). The second, optimization-based group learn an optimization algorithm that is shared across tasks, which can be adapted to new tasks for efficient and effective learning (Ravi & Larochelle, 2017; Andrychowicz et al., 2016; Finn et al., 2017; 2018; Grant et al., 2018; Triantafillou et al., 2017; Rusu et al., 2019; Zhen et al., 2020b). The memory-based meta-learning group leverages an external memory module to store prior knowledge that enables quick adaptation to new tasks (Santoro et al., 2016; Munkhdalai & Yu, 2017; Munkhdalai et al., 2018; Mishra et al., 2018; Zhen et al., 2020a). Our method belongs to the first and third group, as it is based on prototypes with external hierarchical memory, with the goal to perform cross-domain few-shot classification.

**Cross-domain few-shot learning** The problem is formally posed by (Tseng et al., 2020), who attack it with feature-wise transformation layers for augmenting mage features using affine transforms to simulate various feature distribution. Then, (Guo et al., 2020) proposed the cross-domain few-shot learning benchmark, which covers several target domains with varying similarities to natural images. (Phoo & Hariharan, 2020) introduce a method, which allows few-shot learners to adapt feature representations to the target domain while retaining class grouping induced by the base classifier. It performed cross-domain low-level feature alignment and also encodes and aligns semantic structures in the shared embedding space across domains. (Wang & Deng, 2021) proposed a method which improved the cross-domain generalization capability in the cross-domain few-shot learning through task augmentation. In contrast, we first address cross-domain few-shot learning by a variational inference approach, which enables us to better handle the prediction uncertainty on the unseen domains. Moreover, we propose a hierarchical variational memory for cross-domain few-shot learning, and to better deploy the hierarchical memory, we introduce a hierarchical variational prototype model to obtain prototypes at different levels.

## 4 EXPERIMENTS

### 4.1 EXPERIMENTAL SETUP

We apply our method to four cross-domain few-shot challenges and two within-domain few-shot image classification benchmarks. Sample images from all datasets are provided in the appendix A.

**Cross-domain datasets** The 5-way 5-shot cross-domain few-shot classification experiments use *mini*Imagenet (Vinyals et al., 2016) as training domain and test on four different domains, *i.e.*, CropDisease (Mohanty et al., 2016) containing plant disease images, EuroSAT (Helber et al., 2019) consisting of a collection of satellite images, ISIC2018 (Tschandl et al., 2018) containing dermoscopic images of skin lesions, and ChestX (Wang et al., 2017), a set of X-ray images. Results on the 5-way 20-shot and 5-way 50-shot are provided in the appendix.

**Within-domain datasets** The traditional few-shot within-domain experiments are conducted on *mini*Imagenet (Vinyals et al., 2016) which consists of 100 randomly chosen classes from ILSVRC-2012 (Russakovsky et al., 2015), and *tiered*Imagenet (Ren et al., 2019) which is composed of 608

Table 1: Benefit of hierarchical variational prototype in (%) on four cross-domain challenges under the 5-way 5-shot setting. Hierarchical variational prototype achieves slighly better or comparable performance to the variational prototype on all domains.

| Method | CropDiseases | EuroSAT | ISIC | ChestX |
|---|---|---|---|---|
| Variational prototype (Zhen et al., 2020a) | $80.72 \pm 0.37$ | $73.21 \pm 0.41$ | $40.53 \pm 0.35$ | $25.03 \pm 0.41$ |
| **Hierarchical variational prototype** | $\mathbf{83.75} \pm 0.33$ | $\mathbf{74.29} \pm 0.42$ | $\mathbf{41.21} \pm 0.33$ | $\mathbf{26.15} \pm 0.45$ |

Table 2: Hierarchical vs. flat variational memory in (%) on four cross-domain challenges under the 5-way 5-shot setting. Hierarchical variational memory is more critical than the flat variational memory for cross-domain few-shot learning.

| Method | CropDiseases | EuroSAT | ISIC | ChestX |
|---|---|---|---|---|
| Flat variational memory (Zhen et al., 2020a) | $79.12 \pm 0.65$ | $72.21 \pm 0.70$ | $38.97 \pm 0.53$ | $22.15 \pm 1.00$ |
| **Hierarchical variational memory** | $\mathbf{87.65} \pm 0.35$ | $\mathbf{74.88} \pm 0.45$ | $\mathbf{42.05} \pm 0.34$ | $\mathbf{27.15} \pm 0.45$ |

classes grouped in 34 high-level categories. In the test stage, we measure the accuracy of 600 tasks sampled from the meta-test set. In this paper, we focus on 5-way 1-shot/5-shot tasks following the prior work (Snell et al., 2017; Finn et al., 2017; Zhen et al., 2020a).

**Implementation details** We extract image features using a ResNet-10 backbone, which is commonly used for cross-domain few-shot classification (Ye et al., 2020; Guo et al., 2020). For the computation of features at each level, we first input the last convolutional layer feature map of each residual block through a flattening operation and then input the flattened features into the two fully connected layers. For the within-domain experiments, we use two backbones: Conv-4 and ResNet-12. The Conv-4 was first used for few-shot classification by (Vinyals et al., 2016), and is widely used (Snell et al., 2017; Finn et al., 2017; Sun et al., 2019; Zhen et al., 2020a). To gain better performance, ResNet-12 (Bertinetto et al., 2019; Gidaris & Komodakis, 2018; Yoon et al., 2018) is also widely reported for few-shot classification. Following the prior works, we configure the ResNet-12 backbone as 4 residual blocks. Each prototype is obtained in the same way as the cross-domain few-shot classification. We use a two-layer inference network for the few-shot cross domain experiments. Following the VSM (Zhen et al., 2020a), we use the same three-layer inference network for the few-shot within domain experiments. Our code will be publicly released. [1]

**Metrics** The average cross-domain/ within-domain few-shot classification accuracy (%, top-1) along with 95% confidence intervals are reported across all test images and tasks.

## 4.2 RESULTS

**Benefit of hierarchical prototypes** To show the benefit of the hierarchical prototypes, we compare hierarchical variational prototypes with variational prototypes (Zhen et al., 2020a), which obtains probabilistic prototypes of the last layer only. Table 1 shows the hierarchical prototype achieves a slightly better or comparable performance to the single prototypes on all cross-domain few-shot classification tasks. More importantly, the experiments confirm that hierarchical prototypes can capture features at different levels, which allows for better deployment of hierarchical memory. As it may fetch the corresponding information for each level of prototype, which we will demonstrate next.

**Hierarchical vs. flat variational memory** We first show that a flat memory as used in (Zhen et al., 2020a) is less suitable for cross-domain few-shot learning in Table 2. With flat memory, performance even degrades a bit compared to variational prototypes (see Table 1). One interesting observation is that on ChestX – with the largest domain gap – the performance of flat memory degrades the most compared to the variational prototype. This is reasonable since features vary for the domains, resulting in the tested domains not finding the appropriate semantic information. Our hierarchical variational memory consistently surpasses flat memory on all test domains. We attribute the improvements to our model's ability to leverage memory of different layers to generate prototypes. The hierarchical memory provides more context information at different layers, enabling more information to be transferred to new domains, and thus leads to improvements over flat memory.

**Benefit of learning to weigh prototypes** We investigate the benefit of adding the learning to weigh prototypes to the hierarchical variational memory. The experimental results are reported in Table 3. Hierarchical variational memory with bagging produces the final result by bagging (Breiman, 1996)

---

[1] https://github.com/YDU-uva/HierMemory.

Table 3: Benefit of learning to weigh prototypes in (%) on four cross-domain challenges. Hierarchical variational memory with learning to weigh prototypes achieves better performance than with bagging.

| Method | CropDiseases | EuroSAT | ISIC | ChestX |
|---|---|---|---|---|
| Hierarchical variational memory with bagging | 85.43 ± 0.33 | 74.02 ± 0.41 | 40.39 ± 0.32 | 25.98 ± 0.43 |
| **Hierarchical variational memory** | **87.65** ± 0.35 | **74.88** ± 0.45 | **42.05** ± 0.34 | **27.15** ± 0.45 |

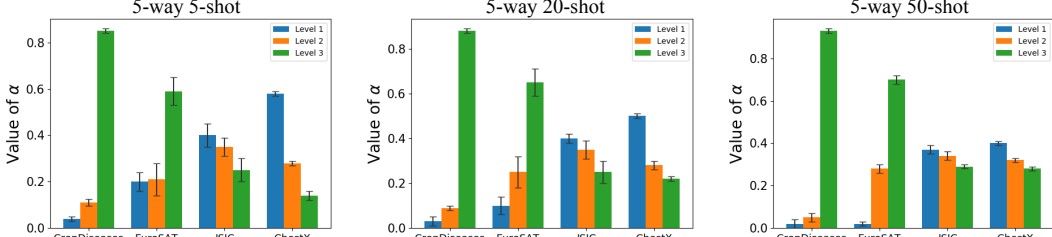

Figure 2: Visualization of weights $\alpha^l$ at different residual blocks. The weight $\alpha^l$ varies per test-domain, the larger the domain shift between the training and test domain, the larger the weight of the low-level prototype.

to vote which category it should be categorized into. Adding the learning to weigh prototypes on top of the hierarchical variational memory leads to a small but consistent gain under all test domains. Thus, the learned adaptive weight acquires the ability to adaptively choose the most generalizable features for the new test domain, resulting in improved performance.

**Importance of prototypes at different levels** We present a visualization of the influence of the prototypes at different levels, using a varying number of shots, in Fig. 2. We sample 600 test tasks on each domain to show the value of $\alpha^l$ at different blocks, and average $\alpha^l$ among all the test tasks. We choose the last three blocks of ResNet-10 to generate the memory and prototype. As presented in (Guo et al., 2020), the degree of domain shift between *mini*ImageNet and target domains is ordered by: CropDiseases < EuroSAT < ISIC < ChestX. From Fig. 2, we can see that the value of $\alpha$ increases along with the increase in domain shift. Another interesting observation is that as the training samples (shots) in each class increase, the weight of the high-level prototypes also increases. This is expected as high-level features can already better represent the information of the new domain due to the increase in the number of training samples. However, the weight of the low-level prototypes is still larger than the weight of the high-level ones. This experiment demonstrates that the lower-level features may be more useful across different domains than the more specialized high-level concepts.

**Visualization of hierarchical variational prototypes** To understand the empirical benefit of using prototype at different levels, we visualize the distribution of prototypes obtained by different blocks on the most challenging ChestX domain under the 5-way 5-shot setting in Fig. 3. The level 1 (low-level) prototypes enable different classes to be more distinctive and distant from each other, while the prototypes obtained by level 3 (high-level) have much more overlap among classes. This suggests that with a large domain shift between the original source domain and the new target domain, low-level textures are more useful and distinctive. This again demonstrates that our hierarchical variational memory to leverage prototypes at different levels is suitable for cross-domain few-shot learning.

Table 4: Comparative results of different algorithms on four proposed cross-domain few-shot challenges. The results of other methods are provided by (Guo et al., 2020). Runner-up method is underlined. Our hierarchical variational memory is a consistent top-performer.

| | CropDiseases 5-way | | EuroSAT 5-way | | ISIC 5-way | | ChestX 5-way | |
|---|---|---|---|---|---|---|---|---|
| Method | 5-shot | 20-shot | 5-shot | 20-shot | 5-shot | 20-shot | 5-shot | 20-shot |
| MatchingNet | 66.39 ± 0.78 | 76.38 ± 0.67 | 64.45 ± 0.63 | 77.10 ± 0.57 | 36.74 ± 0.53 | 45.72 ± 0.53 | 22.40 ± 0.70 | 23.61 ± 0.86 |
| MatchingNet+FWT | 62.74 ± 0.90 | 74.90 ± 0.71 | 56.04 ± 0.65 | 63.38 ± 0.69 | 30.40 ± 0.48 | 32.01 ± 0.48 | 21.26 ± 0.31 | 23.23 ± 0.37 |
| MAML | 78.05 ± 0.68 | 89.75 ± 0.42 | 71.70 ± 0.72 | 81.95 ± 0.55 | 40.13 ± 0.58 | 52.36 ± 0.57 | 23.48 ± 0.96 | 27.53 ± 0.43 |
| ProtoNet | 79.72 ± 0.67 | 88.15 ± 0.51 | 73.29 ± 0.71 | 82.27 ± 0.57 | 39.57 ± 0.57 | 49.50 ± 0.55 | 24.05 ± 1.01 | 28.21 ± 1.15 |
| ProtoNet+FWT | 72.72 ± 0.70 | 85.82 ± 0.51 | 67.34 ± 0.76 | 75.74 ± 0.70 | 38.87 ± 0.52 | 43.78 ± 0.47 | 23.77 ± 0.42 | 26.87 ± 0.43 |
| RelationNet | 68.99 ± 0.75 | 80.45 ± 0.64 | 61.31 ± 0.72 | 74.43 ± 0.66 | 39.41 ± 0.58 | 41.77 ± 0.49 | 22.96 ± 0.88 | 26.63 ± 0.92 |
| RelationNet+FWT | 64.91 ± 0.79 | 78.43 ± 0.59 | 61.16 ± 0.70 | 69.40 ± 0.64 | 35.54 ± 0.55 | 43.31 ± 0.51 | 22.74 ± 0.40 | 26.75 ± 0.41 |
| MetaOpt | 68.41 ± 0.73 | 82.89 ± 0.54 | 64.44 ± 0.73 | 79.19 ± 0.62 | 36.28 ± 0.50 | 49.42 ± 0.60 | 22.53 ± 0.91 | 25.53 ± 1.02 |
| **Ours** | **87.65** ± 0.35 | **95.13** ± 0.35 | **74.88** ± 0.45 | **84.81** ± 0.34 | **42.05** ± 0.34 | **54.97** ± 0.35 | **27.15** ± 0.45 | **30.54** ± 0.47 |

Level 1           Level 2           Level 3

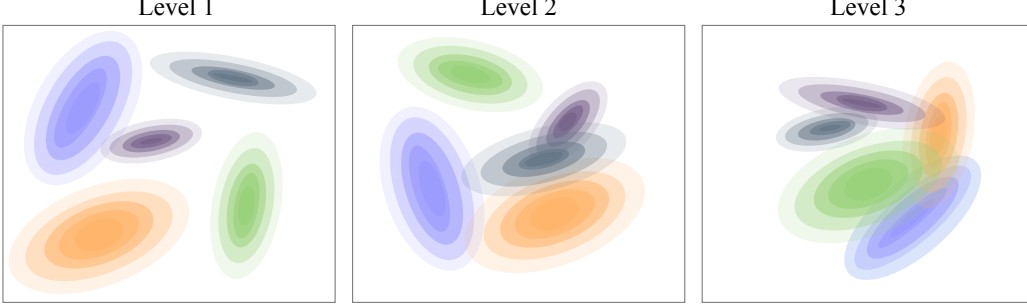

Figure 3: Prototype distributions at different levels with our hierarchical variational memory on ChestX under the 5-way 5-shot setting, where different colors indicate different categories. The prototypes of Level 1 are more distinctive and distant from each other, which reflects the lower-level feature are more crucial when the domain shift is large.

**Few-shot across domains** We evaluate hierarchical variational memory on the four different datasets under 5-way 5-shot and 5-way 20-shot in Table 4, the results for 50-shot are provided in the appendix B.4. Our hierarchical variational memory achieves state-of-the-artperformance on all four cross-domain few-shot learning benchmarks under each setting. On CropDisease (Mohanty et al., 2016), our model achieves high recognition accuracy under various shot configurations, surpassing the second best method, *i.e.* ProtoNet (Snell et al., 2017), by a large margin of 7.93% on the 5-way 5-shot. Even on the most challenging ChestX (Wang et al., 2017), which has a large domain gap with the *mini*ImageNet, delivers 27.15% on the 5-way 5-shot setting, surpassing the second best ProtoNet (Snell et al., 2017) by 3.10%. The consistent improvements on all benchmarks under various configurations confirm that our hierarchical variational memory is effective for cross-domain few-shot learning.

**Few-shot within domain** We also evaluate our method on few-shot classification within domains, in which the training domain is consistent with the test domain. The results using ResNet-12 are reported in Table 5, the results using Conv-4 are reported in the appendix B.5. Our hierarchical variational memory achieves consistently better performance compared to the previous methods on both datasets. The results demonstrate that hierarchical variational memory also benefits performance for few-shot learning within domains.

Table 5: Comparative results for few-shot learning on *mini*Imagenet and *tiered*Imagenet using a ResNet-12 backbone. Runner-up method is underlined. The proposed hierarchical variational memory can also improve performance for few-shot learning within domains.

| Method | *mini*Imagenet 5-way | | *tiered*Imagenet 5-way | |
|---|---|---|---|---|
| | 1-shot | 5-shot | 1-shot | 5-shot |
| SNAIL (Mishra et al., 2018) | $55.71 \pm 0.99$ | $68.88 \pm 0.92$ | - | - |
| Dynamic FS (Gidaris & Komodakis, 2018) | $55.45 \pm 0.89$ | $70.13 \pm 0.68$ | - | - |
| TADAM (Oreshkin et al., 2018) | $58.50 \pm 0.30$ | $76.70 \pm 0.30$ | - | - |
| MTL (Sun et al., 2019) | $61.20 \pm 1.80$ | $75.50 \pm 0.80$ | - | - |
| VariationalFSL (Zhang et al., 2019) | $61.23 \pm 0.26$ | $77.69 \pm 0.17$ | - | - |
| TapNet (Yoon et al., 2019) | $61.65 \pm 0.15$ | $76.36 \pm 0.10$ | $63.08 \pm 0.15$ | $80.26 \pm 0.12$ |
| MetaOptNet (Lee et al., 2019) | $62.64 \pm 0.61$ | $78.63 \pm 0.46$ | $65.81 \pm 0.74$ | $81.75 \pm 0.53$ |
| CTM (Li et al., 2019) | $62.05 \pm 0.55$ | $78.63 \pm 0.06$ | $64.78 \pm 0.11$ | $81.05 \pm 0.52$ |
| CAN (Hou et al., 2020) | $63.85 \pm 0.48$ | $79.44 \pm 0.34$ | $69.89 \pm 0.51$ | $84.23 \pm 0.37$ |
| VSM (Zhen et al., 2020a) | $\underline{65.72} \pm 0.57$ | $\underline{82.73} \pm 0.51$ | $\underline{72.01} \pm 0.71$ | $\underline{86.77} \pm 0.44$ |
| **Ours** | $\mathbf{67.83} \pm 0.32$ | $\mathbf{83.88} \pm 0.25$ | $\mathbf{73.67} \pm 0.34$ | $\mathbf{88.05} \pm 0.14$ |

## 5 CONCLUSION

In this paper, we propose hierarchical variational memory for few-shot learning across domains. The hierarchical memory stores features at different levels, which is incorporated as an external memory into the hierarchical variational prototype model to obtain prototypes at different levels. Furthermore, to explore the importance of different feature levels, we propose learning to weigh prototypes in a data-driven way, which further improves generalization performance. Extensive experiments on six benchmarks demonstrate the efficacy of each component in our model and the effectiveness of hierarchical variational memory in handling both the domain shift and few-shot learning problems.

## 6 ACKNOWLEDGEMENT

This work is financially supported by the Inception Institute of Artificial Intelligence, the University of Amsterdam and the allowance Top consortia for Knowledge and Innovation (TKIs) from the Netherlands Ministry of Economic Affairs and Climate Policy.

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

# A  DATASETS

We apply our method to four cross-domain few-shot challenges and two within-domain few-shot image classification benchmarks. Sample images from each dataset are provided in Figure 4.

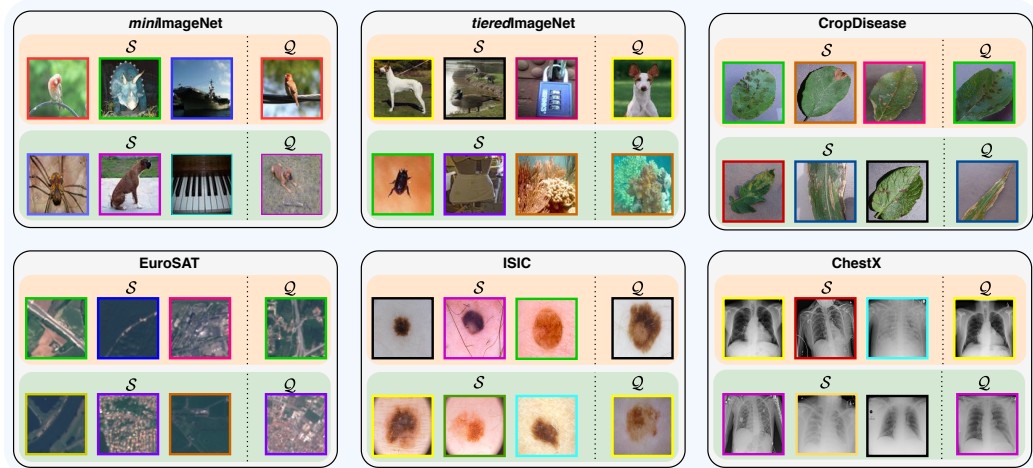

Figure 4: Examples from each dataset. The color of the border of each image represents the category of this image in this task. $\mathcal{S}$ and $\mathcal{Q}$ indicate the support and query sets for each task.

# B  MORE RESULTS

The more ablation results on the 5-way 20-shot and 5-way 50-shot for cross-domain few-shot classification are shown in Sec. B.1, Sec. B.2 and Sec. B.3. The results on four proposed cross-domain few-shot challenges under 5-way 50-shot setting are shown in Sec. B.4. The results of few-shot learning methods on the two benchmark datasets *mini*Imagenet and *tiered*Imagenet by the Conv-4 are shown in the Sec. B.5.

## B.1  BENEFIT OF HIERARCHICAL PROTOTYPE

To show the benefit of the hierarchical prototypes, we compare hierarchical variational prototypes with variational prototypes (Zhen et al., 2020a), which obtains the probabilistic prototypes of the last layer. Table 10 and Table 11 show the the hierarchical variational prototypes consistently outperform variational prototypes on all cross-domain few-shot classification tasks under the 5-way 20-shot and 5-way 50-shot setting.

Table 6: Benefit of hierarchical variational prototype in (%) on four cross-domain challenges under 5-way 20-shots setting.

| Method | CropDiseases | EuroSAT | ISIC | ChestX |
|---|---|---|---|---|
| Variational prototype (Zhen et al., 2020a) | 89.73 $\pm$ 0.35 | 83.21 $\pm$ 0.37 | 50.64 $\pm$ 0.38 | 29.12 $\pm$ 0.46 |
| **Hierarchical variational prototype** | **93.15** $\pm$ 0.33 | **85.13** $\pm$ 0.34 | **52.65** $\pm$ 0.39 | **30.14** $\pm$ 0.45 |

Table 7: Benefit of hierarchical variational prototype in (%) on four cross-domain challenges under 5-way 50-shots setting.

| Method | CropDiseases | EuroSAT | ISIC | ChestX |
|---|---|---|---|---|
| Variational prototype (Zhen et al., 2020a) | 92.01 $\pm$ 0.33 | 81.96 $\pm$ 0.37 | 54.56 $\pm$ 0.32 | 30.98 $\pm$ 0.42 |
| **Hierarchical variational prototype** | **94.25** $\pm$ 0.31 | **84.33** $\pm$ 0.35 | **57.74** $\pm$ 0.32 | **31.27** $\pm$ 0.43 |

## B.2 HIERARCHICAL VS. FLAT VARIATIONAL MEMORY

We show that a flat memory as used in (Zhen et al., 2020a) is less suitable for cross-domain few-shot learning in Table 8 and 9 under the 5-way 20-shot and 5-way 50-shot setting. Hierarchical variational memory consistently achieves the best performance among all the datasets and settings.

Table 8: Hierarchical vs. flat variational memory in (%) on four cross-domain challenges under 5-way 20-shot setting.

| Method | CropDiseases | EuroSAT | ISIC | ChestX |
|---|---|---|---|---|
| Flat variational memory (Zhen et al., 2020a) | 87.27 ± 0.33 | 81.25 ± 0.37 | 49.10 ± 0.35 | 26.21 ± 0.45 |
| **Hierarchical variational memory** | **95.13** ± 0.35 | **84.81** ± 0.34 | **54.97** ± 0.35 | **30.54** ± 0.47 |

Table 9: Hierarchical vs. flat variational memory in (%) on four cross-domain challenges under 5-way 50-shot setting.

| Method | CropDiseases | EuroSAT | ISIC | ChestX |
|---|---|---|---|---|
| Flat variational memory (Zhen et al., 2020a) | 91.59 ± 0.32 | 80.28 ± 0.35 | 50.15 ± 0.31 | 28.12 ± 0.42 |
| **Hierarchical variational memory** | **97.83** ± 0.33 | **87.16** ± 0.35 | **61.71** ± 0.32 | **32.76** ± 0.46 |

## B.3 BENEFIT OF LEARNING TO WEIGH PROTOTYPES

Table 10 and 11 show the results of hierarchical variational memory with bagging vs. hierarchical variational memory for the cross-domain few-shot classification under the 5-way 20-shot and 5-way 50-shot setting. hierarchical variational memory with learning to weigh prototype performs best overall.

Table 10: Benefit of learning to weigh prototypes in (%) on four cross-domain challenges under 5-way 20-shot setting.

| Method | CropDiseases | EuroSAT | ISIC | ChestX |
|---|---|---|---|---|
| Hierarchical variational memory with bagging | 93.39 ± 0.34 | 83.35 ± 0.35 | 53.13 ± 0.38 | 29.17 ± 0.45 |
| **Hierarchical variational memory** | **95.13** ± 0.35 | **84.81** ± 0.34 | **54.97** ± 0.35 | **30.54** ± 0.47 |

## B.4 FEW-SHOT ACROSS DOMAIN

We evaluate hierarchical variational memory on the four different datasets under 5-way and 50-shot configurations in Table 12. Our hierarchical variational memory achieves the new state-of-the-art performance on all the cross-domain few-shot learning benchmarks under 5-way and 50-shot setting.

## B.5 FEW-SHOT WITHIN DOMAIN

We report the results of few-shot learning methods on the two few-shot within domain benchmark datasets *mini*Imagenet and *tiered*Imagenet by the Conv-4 backbones in Table 13. Under both datasets, hierarchical variational memory consistently outperforms the previous approaches.

## C TRAINING SPEED

We plot the training loss versus training iterations for different algorithms in Figure 5. From Figure 5 and Table 4, we conclude our hierarchical variational memory achieves best training efficiency and classification accuracy, which demonstrates that our hierarchical variational memory is effective for cross-domain and within domain few-shot learning.

## D RESULTS ON DATA AUGMENTATION

We also provide results for few-shot within domain using a ResNet-12 backbone under data augmentation in the meta-training stage following (Zhang et al., 2021). The results are shown in Table 14. With

Table 11: Benefit of learning to weigh prototypes in (%) on four cross-domain challenges under 5-way 50-shot setting.

| Method | CropDiseases | EuroSAT | ISIC | ChestX |
|---|---|---|---|---|
| Hierarchical variational memory with bagging | 95.91 ± 0.35 | 85.90 ± 0.34 | 58.97 ± 0.33 | 30.95 ± 0.43 |
| **Hierarchical variational memory** | **97.83** ± 0.33 | **87.16** ± 0.35 | **61.71** ± 0.32 | **32.76** ± 0.46 |

Table 12: Comparative results of few-shot learning methods on four proposed cross-domain few-shot challenges under 5-way 50-shot setting.

| Method | CropDiseases | EuroSAT | ISIC | ChestX |
|---|---|---|---|---|
| MatchingNet | 58.53 ± 0.73 | 54.44 ± 0.67 | 54.58 ± 0.65 | 22.12 ± 0.88 |
| MatchingNet+FWT | 75.68 ± 0.78 | 62.75 ± 0.76 | 33.17 ± 0.43 | 23.01 ± 0.34 |
| ProtoNet | 90.81 ± 0.43 | 80.48 ± 0.57 | 41.05 ± 0.52 | 29.32 ± 1.12 |
| ProtoNet+FWT | 87.17 ± 0.50 | 78.64 ± 0.57 | 49.84 ± 0.51 | 30.12 ± 0.46 |
| RelationNet | 85.08 ± 0.53 | 74.91 ± 0.58 | 49.32 ± 0.51 | 28.45 ± 1.20 |
| RelationNet+FWT | 81.14 ± 0.56 | 73.84 ± 0.60 | 46.38 ± 0.53 | 27.56 ± 0.40 |
| MetaOpt | 91.76 ± 0.38 | 83.62 ± 0.68 | 54.80 ± 0.54 | 29.35 ± 0.99 |
| **Ours** | **97.83** ± 0.33 | **87.16** ± 0.35 | **61.71** ± 0.32 | **32.76** ± 0.46 |

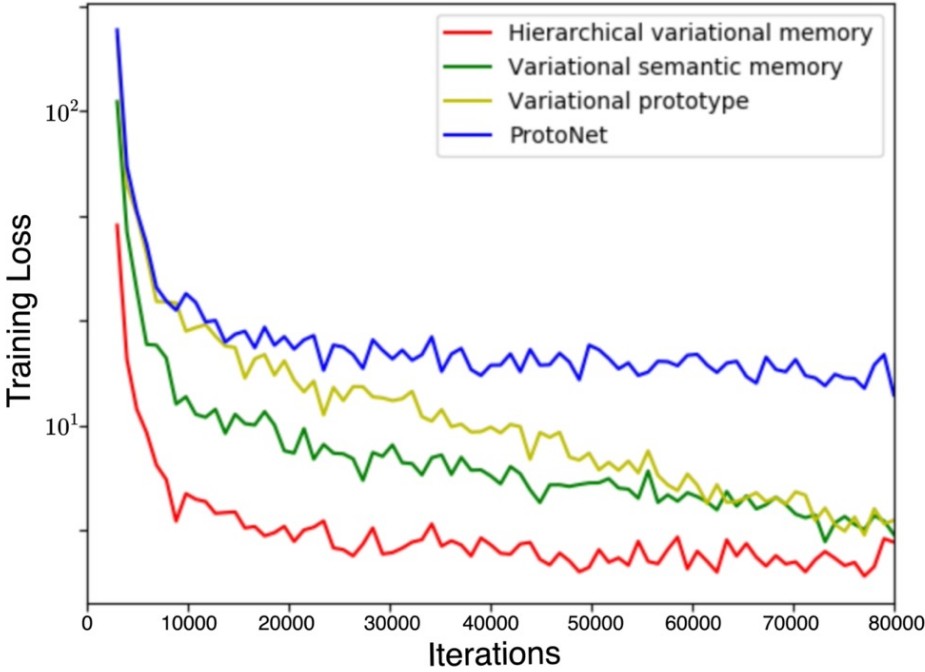

Figure 5: Training loss versus iterations for different algorithms. Our hierarchical variational memory achieves fastest training convergence.

data augmentation for few-shot within domain, our model also consistently achieves thecompetitive performance.

# E  BENEFITS OF HIERARCHICAL STRUCTURES

We first show the benefit of hierarchical structures for few-shot within domain in Table 15. To show the effect of our hierarchical formulation, we compare with VSM, which does not use the hierarchical structure for the prototype, and with hierarchical variational memory (last level). We found a performance improvement by using the hierarchical formulation for the prototype. Also, by

Table 13: Comparative results of few-shot learning methods on *mini*Imagenet and *tiered*Imagenet using a Conv-4 backbone. Runner-up method is underlined.

| Method | *mini*Imagenet 5-way | | *tiered*Imagenet 5-way | |
| --- | --- | --- | --- | --- |
| | 1-shot | 5-shot | 1-shot | 5-shot |
| MatchingNet (Vinyals et al., 2016) | 43.56±0.84 | 55.31±0.73 | - | - |
| Meta-LSTM (Ravi & Larochelle, 2017) | 43.44±0.77 | 60.60±0.71 | - | - |
| MAML (Finn et al., 2017) | 48.70±1.84 | 63.11±0.92 | 51.67±1.81 | 70.30±1.75 |
| ProtoNets (Snell et al., 2017) | 49.42±0.78 | 68.20±0.66 | 48.58±0.87 | 69.57±0.75 |
| Reptile (Nichol et al., 2018) | 47.07±0.26 | 62.74±0.37 | 48.97±0.21 | 66.47±0.21 |
| RelationNet (Sung et al., 2018) | 50.44±0.82 | 65.32±0.70 | 54.48±0.93 | 71.32±0.78 |
| IMP (Allen et al., 2019) | 49.60±0.80 | 68.10±0.80 | - | - |
| FEAT (Ye et al., 2020) | 55.15±0.20 | 71.61±0.16 | - | - |
| VSM (Zhen et al., 2020a) | 54.73±1.60 | 68.01±0.90 | 56.88 ± 1.71 | 74.65 ± 0.81 |
| **Ours** | **57.04**±0.92 | **72.65**±0.20 | **59.01**±0.83 | **77.76**±0.62 |

Table 14: Comparative results for few-shot learning on *mini*Imagenet and *tiered*Imagenet using a ResNet-12 backbone under same data augmentation as Zhang et al. (2021). With data augmentation, our model also achieves competitive performance.

| Method | *mini*Imagenet 5-way | | *tiered*Imagenet 5-way | |
| --- | --- | --- | --- | --- |
| | 1-shot | 5-shot | 1-shot | 5-shot |
| Zhang et al. (2021) | 69.68 ± 0.76 | 81.65 ± 0.54 | 74.19 ± 0.90 | 86.09 ± 0.60 |
| **Ours** | **71.05** ± 0.31 | **84.54** ± 0.23 | **75.43** ± 0.29 | **88.97** ± 0.24 |

comparing the hierarchical variational memory (last level) and hierarchical variational memory, we found that the performance has further improved, which shows the effect of ensembling.

## F BENEFITS OF MODEL SIZE

We also ablate our approach in terms of model size, by varying the backbone. We report few-shot classification within domain by using Conv-4, WRN-28-10, ResNet-12 and ResNet-18 in Table 16. The performance of our method increases along with the increase in backbone capacity. As our memory is much smaller than the backbones, the trend in experimental results is not affected by the change of backbone.

Table 15: Benefit of learning hierarchical structure for few-shot learning using a Conv-4 backbone. The hierarchical structure benefits performance compared to the flat VSM.

| Method | *mini*Imagenet 5-way | | *tiered*Imagenet 5-way | |
| --- | --- | --- | --- | --- |
| | 1-shot | 5-shot | 1-shot | 5-shot |
| VSM (Zhen et al., 2020a) | 54.73±1.60 | 68.01±0.90 | 56.88 ± 1.71 | 74.65 ± 0.81 |
| Hierarchical variational memory (last level) | 56.72±0.90 | 70.11±0.21 | 57.89 ± 0.81 | 76.01 ± 0.60 |
| **Hierarchical variational memory** | **57.04**±0.92 | **72.65**±0.20 | **59.01**±0.83 | **77.76**±0.62 |

Table 16: Comparative results for few-shot learning on *mini*Imagenet and *tiered*Imagenet using different backbones.

| Backbone | *mini*Imagenet 5-way | | *tiered*Imagenet 5-way | |
| --- | --- | --- | --- | --- |
| | 1-shot | 5-shot | 1-shot | 5-shot |
| Conv-4 | 57.04 ± 0.92 | 72.65 ± 0.20 | 59.01 ± 0.83 | 77.76 ± 0.62 |
| WRN-28-10 | 66.19 ± 0.35 | 81.61 ± 0.20 | 69.81 ± 0.33 | 84.76 ± 0.12 |
| ResNet-12 | 67.01 ± 0.32 | 81.75 ± 0.21 | 71.70 ± 0.35 | **85.13** ± 0.09 |
| ResNet-18 | **68.94** ± 0.30 | **82.75** ± 0.25 | **73.13** ± 0.88 | 85.06 ± 0.14 |

