# OpenReview forum: "Hierarchical Variational Memory for Few-shot Learning Across Domains"
_ICLR.cc/2022/Conference — ICLR 2022 Poster_

### Official Review · Reviewer_g1Bf · 2021-11-03

**Correctness:** 3
**Technical Novelty And Significance:** 3
**Empirical Novelty And Significance:** Not applicable
**Recommendation:** 8
**Confidence:** 4

**Main Review:**

$\textbf{Strengths}$
1. The paper is well written and easy to follow. The authors do a good job of introducing their model elements, and contrasting them with previous works.
2. The improvements obtained over the baselines are impressive. Specifically, on the task of cross-domain few shot classification, the gap between the proposed method and the most competitive baseline is significant.
3. Additionally, the ablation experiments shown are extensive and do a good job of highlighting the importance of each component in the proposed framework.

$\textbf{Weaknesses}$
1. Table 13 highlights that the proposed method is not that effective when using shallow feature extractors like Conv-4. This probably leads me to believe that ensembling is providing the major improvements and the proposed hierarchical formulation isn't actually that important. To this end, I have two questions:

a. What happens if you don't use an ensemble and rather just use the logits from the last level (keeping the rest of the architecture as is)? As the later latent variables depend on the earlier ones (Figure 1), if the hierarchical framework works well, you should still see improvements over the baselines.

b. What happens if you train $L$ instances of VSM (Zhen et al. 2020), where each instance is trained on the output of a residual block. That is, if you remove the connections between $\mathbf{z}$ and $\mathbf{m}$ in Figure 1. How crucial is it to have these dependencies between the latent variables.
It would be great if the authors could comment on this.

2. What is the memory overhead of using the proposed hierarchical method over the baseline (Zhen et al. 2020)? I can imagine having $L$ layers of memory to considerably increase the memory overhead. Additionally, due to the increase in the number of latent variables, is convergence slower as well?

3. The VSM numbers shown in Table 5 are lower than what is reported in (Zhen et al. 2020). The numbers in (Zhen et al. 2020) show that the proposed method is inferior when compared to VSM on within-domain few shot classification. I understand that the authors re-implemented their model, but is there an explanation as to why the reimplemented numbers are considerably lower?



**Summary Of The Paper:**

The authors propose a novel model that focuses on improving cross domain few shot classification. To this end they introduce a hierarchical extension to the work proposed in (Zhen et al. 2020), wherein the latent variables at different levels capture distinct semantic information. The proposed framework enables generating class specific prototypes at different hierarchical levels, which are then used to make predictions at each level. These predictions are ensembled using domain specific weights obtained from the support set via a gradient based method. Through experiments on various cross-domain and in-domain tasks, the authors show considerable improvements over the baselines.

**Summary Of The Review:**

This work provides a logical extension to the existing work in (Zhen et al. 2020) by introducing a hierarchical variational memory framework. Through the experiment results it is evident that the proposed method provides considerable improvements over existing approaches. I have some concerns regarding the actual importance of dependencies within the latent variables. I'm still inclined to accept this paper, and would be willing to increase my rating if the authors address my concerns.

---

> ### Author Response · Authors · 2021-11-20
> **Response to Reviewer g1Bf**
>
> **What happens if you don't use an ensemble and rather just use the logits from the last level (keeping the rest of the architecture as is)?**
>
> Thank you for sharing the insight. To show the importance of our hierarchical formulation,  we followed your suggestion to add experiments that just use the logits from the last level. The results are shown in the table below,
>
> |                                                    | miniImagenet | miniImagenet | tieredImagenet | tieredImagenet |
> | -------------------------------------------------- | ------------ | ------------ | -------------- | -------------- |
> |                                                    | 1-shot       | 5-shot       | 1-shot         | 5-shot         |
> | VSM                                                | 54.73 ± 1.60 | 68.01 ± 0.90 | 56.88 ± 1.71  | 74.65 ± 0.81  |
> | Hierarchical variational memory (last level only)| 56.72 ± 0.90 | 70.11 ± 0.21 | 57.89 ± 0.81  | 76.01 ± 0.60  |
> | Hierarchical variational memory                    | 57.04 ± 0.92 | 72.65 ± 0.20 | 59.01 ± 0.83  | 77.76 ± 0.62  |
>
> Comparing VSM which do not use a hierarchical formulation for the memory and  Hierarchical variational memory (last level only), shows the importance of a hierarchical formulation.
> Comparing Hierarchical variational memory (last level only) and Hierarchical variational memory shows the effect of ensembling.
> We have added the results and analysis on the benefits of the hierarchical structure for few-shot within domain in Appendix Table 16.
>
> **What happens if you train  L  instances of VSM (Zhen et al. 2020), where each instance is trained on the output of a residual block. That is, if you remove the connections between z and m in Figure 1. How crucial is it to have these dependencies between the latent variables. It would be great if the authors could comment on this.**
>
> The generation of latent prototype **z** must be conditional on **m** in our hierarchical variational memory. If we remove the connections between **z** and **m** in Figure 1, our hierarchical variational memory can no longer use the memory and then it actually corresponds to the hierarchical variational prototype model.  By comparing the hierarchical variational prototype results in Table 1 (e.g., 83.75% in CropDiseases) and the hierarchical variational memory in Table 2  (e.g., 87.65% in CropDiseases), the benefits of using connections between **z** and **m** can be shown. We will better highlight this benefit in the main paper.
>
> **What is the memory overhead of using the proposed hierarchical method over the baseline (Zhen et al. 2020)? I can imagine having L  layers of memory to considerably increase the memory overhead. Additionally, due to the increase in the number of latent variables, is convergence slower as well?**
>
> Compared to VSM we have two more layers of memory, which doubles the memory size from 4K to about 8K. We have added a convergence analysis and plot the training loss versus iterations for the ProtoNets, variational prototype network, variational semantic memory and our model in Appendix C and Figure 4.  Although memory size increases, the convergence of our hierarchical variational memory is still faster than VSM. The first reason is that our memory size (8K) is negligible compared to the backbone capacity (10M). Another reason is that our model adaptively chooses the most generalizable features per training task, which speeds up the training.
>
> **The VSM numbers shown in Table 5 are lower than what is reported in (Zhen et al. 2020). $\cdots$ I understand that the authors re-implemented their model, but is there an explanation as to why the reimplemented numbers are considerably lower?**
>
> The reviewer is right. We provide the explanation and repairment in our response to the same observation by Reviewer YEXZ above. Thank you.

---

> > ### Comment · Reviewer_g1Bf · 2021-11-29
> > **Thanks for the clarifications.**
> >
> > I would like to thank the authors for answering all my questions and updating the manuscript with the updated experiments. I will update my initial rating to an accept.

---

### Official Review · Reviewer_YEXZ · 2021-11-04

**Correctness:** 3
**Technical Novelty And Significance:** 2
**Empirical Novelty And Significance:** 2
**Recommendation:** 5
**Confidence:** 4

**Main Review:**

From a technical standpoint, the work appears to be a relatively straightforward extension of [Zhen et al., 2020].  Hence, experimental validation of the impact of the proposed per-layer memory model is especially important.  The paper presents results on the same few-shot tasks as Zhen et al. (mini-Imagenet and tiered-Imagenet), as well as comparison to other meta-learning methods on few-shot cross-domain tasks.  Here, ablation experiments also show learned weighting of per-layer prototypes to be useful.

However, the experimental results leave open a critical question about the comparison of the proposed approach to the baseline variational semantic memory (VSM) of Zhen et al.  Specifically, the results quoted for VSM in Table 5 are worse than the results in [Zhen et al., 2020] for this same experiment.  In fact, the results reported in Table 6 of [Zhen et al., 2020] are better in 3 out of 4 settings than the results reported for the proposed system.  The discrepancy is:

Method mini-ImageNet (1-shot/5-shot) tiered-ImageNet (1-shot/5-shot)

VSM  65.72 82.73 72.01 86.77

VSM* 64.21 79.69 69.58 83.28

Ours 67.01 81.75 71.70 85.13

where VSM is as reported in [Zhen et al., 2020], VSM* is the "re-implementation" by this submission, and Ours is the submission's approach.  Since the proposed system is an extension of Zhen et al., it is quite detrimental to actually perform worse than the baseline.  These results are also on a central experimental setting (few-show learning with deep models).  It is not acceptable to present a "re-implementation" that flips the ranking of the methods; at minimum, some extensive explanation is required about differences between the original and "re-implementation" as well as why the original published results were not replicated.


**Summary Of The Paper:**

This paper presents a hierarchical version of the variational memory approach of [Zhen et al., 2020] for few-shot learning.  The core technical methodology follows that developed by Zhen et al., with the difference being in terms of the model: this work utilizes a deep model with per-layer memory, whereas Zhen et al. utilize a single memory for high-level concepts (i.e., deepest layer only).  Motivating the choice of per-layer memory is the desire to better handle few-shot learning tasks with domain shift, as the representations in earlier layers may be more relevant in such scenarios; the proposed framework learns a hypernetwork to predict attention weights over the per-layer prototypes.


**Summary Of The Review:**

While this work appears to be a promising extension of the variational semantic memory of [Zhen et al., 2020] to models with multi-level prototypes, it omits proper comparison to [Zhen et al., 2020] on a central experiment, which, when included shows the proposed approach produces worse performance than the baseline.  The author response should address this discrepancy.

---

> ### Author Response · Authors · 2021-11-20
> **Response to Reviewer YEXZ**
>
> The reviewer is right and we regret our table does not report the original results using the same ResNet-12 as VSM by Zhen et al. (2020). We hope it is still mendable with the new experiments. To keep the inference network of our few-shot cross-domain setting consistent with our few-shot within domain setting, we reported results with only two layers in our inference network and our re-implementation of VSM. However, in their paper Zhen et al. use three layers in their VSM inference network. For a more fair comparison, we  report  the results of our approach using the same three-layer inference network as the original VSM:
>
> |                         | miniImagenet | miniImagenet | tieredImagenet | tieredImagenet |
> | ----------------------- | ------------ | ------------ | -------------- | -------------- |
> | Method                  | 1-shot       | 5-shot       | 1-shot         | 5-shot         |
> | VSM (Zhen et al., 2020) | 65.72 ± 0.57 | 82.73 ± 0.51 | 72.01 ± 0.71  | 86.77 ± 0.44  |
> | Ours                    | 67.83 ± 0.32 | 83.88 ± 0.25 | 73.67 ± 0.34  | 88.05 ± 0.14  |
>
> We have updated the new results in Table 5 and Table 13 and clarified the subtle implementation differences in the experiments. Thank you.

---

### Official Review · Reviewer_2Ajk · 2021-11-04

**Correctness:** 3
**Technical Novelty And Significance:** 3
**Empirical Novelty And Significance:** 3
**Recommendation:** 6
**Confidence:** 4

**Main Review:**

Pros
- The paper is well-written and easy to follow. The paper structure is very clear.
- The idea of leveraging multi-level semantic information is interesting for few-shot learning across domains.  Although using multi-level features is not something new in the space in general, using it as a condition for the prototypes is novel for few-shot learning.  It seems to be a good extension to the previous memory networks.

Concerns
- Would the proposed method with external memory significantly increase the model size?  The external memory might be something unfair to approaches without explicit external memory.
- The baselines in the result tables (Table 4 and 5) may not be the current SODA methods. According to the public leader board https://paperswithcode.com/sota/few-shot-image-classification-on-mini-2. the `prototype completion for few-shot learning` (Zhang et al.) work achieves 69.68% +- 0.76% on the 5-way 1-shot experiment using ResNet-12, which is higher than the reported numbers in Table 5.
- More ablation study in terms of model size (e.g., different backbones), training efficiency (e.g., how fast is the variational inference-based approach?).




**Summary Of The Paper:**

This paper proposes a hierarchical memory to store features at different semantic levels for few-shot learning across domains. It introduces a hierarchical prototype model, where each level of the prototypes fetches the corresponding information from the hierarchical memory.  The authors follow the hyper network design to learn the weights when combining predictions from multiple levels.  The overall model is optimized using a variational inference framework.

The proposed method is evaluated on 4 various datasets which have a domain gap between the training data.  The authors also show that this method is competitive on the commonly used few-shot image classification benchmarks.



**Summary Of The Review:**

The overall idea is interesting but there are concerns regarding the experiments as well as the introduction of external memory.


=== Post-rebuttal Comments===
The authors addressed most of my concerns in the feedback.  I kept my score and learned towards acceptance.

---

> ### Author Response · Authors · 2021-11-20
> **Response to Reviewer 2Ajk**
>
> **Would the proposed method with external memory significantly increase the model size? The external memory might be something unfair to approaches without explicit external memory.**
>
> Our memory only stores the prototype per category, as a 256-dimensional vector. It amounts to a total model-memory size of about 8K. We consider it negligible compared to the 10M parameters of our ResNet-12 backbone. We have clarified the memory size in the implementation details.
>
> **The baselines in the result tables (Table 4 and 5) may not be the current SODA methods. According to the public leader board https://paperswithcode.com/sota/few-shot-image-classification-on-mini-2. the prototype completion for few-shot learning (Zhang et al.) work achieves 69.68% +- 0.76% on the 5-way 1-shot experiment using ResNet-12, which is higher than the reported numbers in Table 5.**
>
> According to their implementation, Zhang et al. [1] use data augmentation while our model does not. For a fair comparison, we added the same data augmentation techniques in our model as well.  The results are:
>
> |                    | miniImagenet | miniImagenet | tieredImagenet | tieredImagenet |
> | ------------------ | ------------ | ------------ | -------------- | -------------- |
> |                    | 1-shot       | 5-shot       | 1-shot         | 5-shot         |
> | Zhang et al. 2021 [1] | 69.68 ± 0.76 | 81.65 ± 0.54 | 74.19 ± 0.90  | 86.09 ± 0.60  |
> | Ours               | 71.05 ± 0.31 | 84.54 ± 0.23 | 75.43 ± 0.29  | 88.97 ± 0.24  |
>
> Our conclusions do not change, our model consistently achieves good performance. The new results with data augmentation have been added in Appendix D Table 14.
>
> Reference:
>
> [1] Zhang et al., Prototype completion for few-shot learning. arXiv preprint arXiv:2108.05010, 2021.
>
> **More ablation study in terms of model size (e.g., different backbones), training efficiency (e.g., how fast is the variational inference-based approach?).**
>
> We added the suggested model size ablation study for within domain few-shot classiﬁcation using WRN-28-10 (about 10M parameters) and ResNet-18 (about 12.6M parameters) in Appendix Table 15. The performance of our method increases along with the increase in backbone capacity.  As our memory is much smaller than the backbone capacity,  the trend in results is not affected.
>
>
> |           | miniImagenet | miniImagenet | tieredImagenet | tieredImagenet |
> | --------- | ------------ | ------------ | -------------- | -------------- |
> | Backbone  | 1-shot       | 5-shot       | 1-shot         | 5-shot         |
> | Conv-4    | 57.04 ± 0.92 | 72.65 ± 0.20 | 59.01 ± 0.83  | 77.76 ± 0.62  |
> | WRN-28-10 | 66.19 ± 0.35 | 81.61 ± 0.20 | 69.81 ± 0.33  | 84.76 ± 0.12  |
> | ResNet-12 | 67.01 ± 0.32 | 81.75 ± 0.21 | 71.70 ± 0.35  | 85.13 ± 0.09  |
> | ResNet-18 | 68.94 ± 0.30 | 82.75 ± 0.25 | 73.13 ± 0.88  | 85.06 ± 0.14  |
>
> We also added the convergence analysis and plot the training loss versus iterations for the ProtoNets, variational prototype network, variational semantic memory and our model in Appendix Figure 5. Our hierarchical variational memory achieves best classiﬁcation accuracy and training efﬁciency. Thank you.

---

### Public Comment · ~Haoqing_Wang1 · 2021-11-19
**Missing important baselines**

Hi, [1] is not considered in you paper, which has been accepted by IJCAI 2021. Importantly, they use a simpler method and achieve high accuracy than yours in 5way-5shot tasks on ChestX, ISIC, EuroSAT and CropDisease. I think you should use [1] as the baseline, or explain the reason why it cannot be used as a baseline.

[1] Cross-Domain Few-Shot Classification via Adversarial Task Augmentation. IJCAI 2021.

---

> ### Author Response · Authors · 2021-11-28
> **Response to Haoqing Wang**
>
> Thank you for the great reference, we will add it in the future version.
>
> Our model is trained from scratch, while [1] uses an additional pre-training strategy which pre-trains the feature extractor on the mini-ImageNet dataset. We will reclarify this point in the future version.

---

### Author Response · Authors · 2021-11-20
**Summary**

We thank all Reviewers for their insightful reviews, sharp comments and supportive suggestions. Please see our point-by-point response to each reviewer comment below.

---

### Decision · Program_Chairs · 2022-01-20

**Decision:**

Accept (Poster)

**Comment:**

This paper presents a hierarchical memory for cross domain and few shot classification problems. The paper is well written, tackles an important topic, and the proposed approach which is an extension of VSM is interesting. Reviewer YEXZ has some concerns regarding comparison to a more proper baseline. I believe that the authors have adequately addressed this. Reviewer 2Ajk and g1Bf also have suggestions that the authors have incorporated in the revision. I recommend accepting this paper.